# Geodiversity and biodiversity on a volcanic island: The role of scattered phonolites for plant diversity and performance

David Kienle[1,A], Anna Walentowitz[1,A,B], Leyla Sungur[1,A], Alessandro Chiarucci[2], Severin D. H. Irl[3], Anke Jentsch[4,5], Ole R. Vetaas[6], Richard Field[7] and Carl Beierkuhnlein[1,5,8]

[1]Biogeography, University of Bayreuth, Bayreuth, 95440, Germany
[2]BIOME Lab, Department of Biological, Geological and Environmental Sciences, Alma Mater Studiorum, University of Bologna, Bologna, 40126, Italy
[3]Biogeography and Biodiversity Lab, Institute of Physical Geography, Goethe-University Frankfurt, Frankfurt, 60438, Germany
[4]Disturbance Ecology, University of Bayreuth, Bayreuth, 95440, Germany
[5]Bayreuth Center of Ecology and Environmental Research BayCEER, University of Bayreuth, Bayreuth, 95440, Germany
[6]Department of Geography, University of Bergen, Bergen, 5020, Norway
[7]School of Geography, University of Nottingham, Nottingham, NG7 2RD, United Kingdom
[8]Geographical Institute Bayreuth, GIB, University of Bayreuth, Bayreuth, 95440, Germany

[A]These authors contributed equally to this study.

[B]*Correspondence to*: Anna Walentowitz (anna.walentowitz@uni-bayreuth.de)

**Abstract.** Oceanic islands are cradles of endemism, contributing substantially to global biodiversity. A similarity in magmatic origin translates into high global comparability of substrates of volcanic islands on the oceanic crust with, however, slightly chemically or physically differentiated petrography in some places. Phonolites are examples of rare localities with intermediate chemical characteristics between felsic and mafic and with diverse textures. They contribute to habitat heterogeneity and offer specific growth conditions in a significantly different matrix of basaltic substrates. The explicit contribution of geodiversity to island biodiversity has been little studied, despite growing evidence of its importance on continents. On the island of La Palma, Canary Islands, isolated phonolitic rocks are conspicuous by their light colour and specific shape. Although these outcrops only cover small areas, their unique form and composition increase within-island geodiversity. To investigate how this affects biodiversity on La Palma, we sampled all vascular plant species in 120 plots on four sets of paired sites, to test if plant diversity and performance is enhanced on phonolitic compared to basaltic rocks. We recorded species number and abundance, individual plant height and diameter as proxies for aboveground resource allocation and tested for differences in vegetation cover and species composition between the bedrock types. We found higher species richness and abundance on phonolites than neighbouring basaltic substrates, and individuals of the same species were larger (in height and diameter) on phonolites compared to neighbouring basalt. An endemic woody species with two distinct varieties even appears almost exclusively on the small surfaces of phonolitic rock. Despite extremely limited spatial extent, phonolitic rocks can play an important role for plant biodiversity on islands.

## 1 Introduction

Biodiversity is known to depend mainly on abiotic drivers, such as climate and topography (Field et al., 2009). However, the importance and explicit impacts of geodiversity on biodiversity have long been insufficiently researched and partly ignored. Only recently, the topic started to receive more attention (e.g., Gray, 2004; Lawler et al., 2015; Bailey et al., 2017; Alahuhta, 2020; Barajas-Barbosa et al., 2020). Geodiversity is in many respects an abiotic equivalent to biodiversity (Gray, 2011) and represents the variability of chemical components, surface structure, edaphic and hydrological features (Gray, 2004; Bailey et

al., 2017). This variability contributes to habitat diversity and thus affects biodiversity patterns via the provisioning of ecological niches (Liu et al., 2013; Gillespie & Roderick, 2014; Bailey et al., 2017). Geological elements provide unique or distinctive habitats for plants and insects, deliver initial growth conditions for vegetation or fungi formation, and are part of nutrient cycling and soil-atmosphere interactions (Tukiainen et al., 2016).

Biodiversity is distributed unevenly throughout the world (Gaston, 2000), to which oceanic islands contribute disproportionately much due to their high endemic richness (Kier et al., 2009). Substrates that differ in geochemistry and petrography are likely to be relevant for biodiversity on oceanic islands, where most rocks commonly share similar volcanic genesis, resulting in only slight differences in the parent material. Distinct substrates with limited extent, such as individual rock types, may function as a second isolating abiotic filter for populations in addition to the spatial isolation of oceanic islands

that are known to be of outstanding importance for speciation at the global scale (Kier et al. 2009). Specific rock habitats, particularly rocks that exhibit petrographic and geochemical substrates such as e.g. serpentinites, are known to be rich in habitat-specific endemics (e.g., Harrison et al., 2006; Kazakou et al., 2010). Those species evolved specific adaptations to the unique nutrient contents and soil conditions and the presence of heavy metals that cannot be tolerated by other plant species (Harrison & Rajakaruna, 2011). This phenomenon is known as well on continents, where substrates such as serpentinite and

gypsum outcrops host specialised floras and contribute to broad-scale diversity (see, e.g., Chiarucci et al., 1998; Pausas et al. 2003). It underlines the relevance of understanding the importance of geodiversity for insular biodiversity, which is particularly vulnerable to extinction due to highly restricted ranges and small population sizes of insular endemic species (Paulay, 1994). Phonolites are rocks that occur at volcanic intraplate settings in insular and continental contexts worldwide (Garcia et al., 1986; Ackerman et al., 2015; Hagos et al., 2017). They exist in a variety of geologic outcrops formed by volcanic activity. Such

outcrops mainly exist on continents, where they are often linked to faults and tectonic activity. Major components of these extrusive igneous rocks (formed from lava with low silica content) are alkaline feldspars together with foid minerals, nepheline, and pyroxene (Ackerman et al., 2015; Abratis et al., 2015) or their conversion products.

On the island of La Palma (Canary Islands, Spain), several phonolitic rock outcrops are embedded into a matrix of basaltic

origin (Middlemost, 1972). The dominant rock type found on La Palma is olivine and augite-titanaugite porphyric basalt, resulting from rapidly rising magma from the upper mantle (Middlemost, 1970). In contrast, there were times when a sizeable

magmatic chamber below the island enabled differentiation of magma and the removal of silica, thus yielding ultramafic, trachytic, and phonolitic rocks (Middlemost, 1970). Phonolite trachytes (showing the exhalation of gases during eruptions) occur on various volcanic islands such as La Palma and St. Helena. On La Palma their distribution is focused on the southern (young) part of the island. The major chemistry of phonolites on La Palma is comparable to that of "average phonolites", as described by Nockolds (1954). The current volcanic activity and lava flow deriving from the Cumbre Vieja volcano (Pankhurst et al., 2021) are a demonstration of how phonolitic rocks became isolated by younger lava solidifying around the peaks of phonolitic rock. This event illustrates that the isolation of the investigated phonolitic habitats (e.g., Roque Teneguía) is far from being a singularity but rather a process that is highly likely to have happened repeatedly on oceanic islands in general.

Volcanic activity with the production of tephra and lava flows is a noticeably young phenomenon in the southern part of La Palma, with eruptions as recent as now (Pankhurst et al. 2021). Thus, the remnant phonolitic rocks are the tips of a former land surface that are today embedded in a basaltic matrix of noticeably immature age (Garantje et al., 1998). In consequence, weathering processes on phonolites were active on longer time scales compared to other surrounding rocks. In addition to petrography, differences in weathering between the rock types and resulting nutrient availability also refer to different time scales of exposure. A higher nutrient availability enables higher plant abundances and larger plant size. Porder et al. (2004) found comparable conditions at a catena of different rock ages in the Hawaiian Islands.

Compared to basaltic lava outcrops, phonolites differ in their chemical composition and additionally, in colour, texture, density, weathering, and formation fracturing (von Fragstein et al., 1988). Tafoni-weathering (Formoso et al., 1989) can be observed on phonolitic surfaces, indicating temperature and moisture gradients between the surface and the solid body of rocks (Brandmeier et al., 2011) that appear in combination with wind exposure. Circulating leachate reaches the rock's surface and evaporates, exposing its dissolved mineral content and enabling the development of secondary mineral assemblages (Spürgin et al., 2019). These can contribute to plant nutrient supply, which is also why ground phonolite rock powder can be used as an effective fertilizer (Faccini et al., 2015). For phonolites, increased release of nutrients can be mediated by bioweathering actions and plants receiving this fertilizer showed higher productivity and increased accumulation of the macronutrient potassium in plants could be detected when applying phonolite rock powder (Tavares et al., 2018; Nogueira et al., 2021). Phonolites and the related larger-grained nepheline syenites contain significantly larger amounts of the essential nutrient potassium (see A1 for a literature overview). Even if quantitatively small, such processes are of particular importance at nutrient-poor sites. In contrast, the young basalts in the southern part of La Palma are barely weathered (Carracedo et al., 1999), appearing rough and friable with sharp spikes. We expect these petrographic and geochemical differences of parent material to affect vegetation cover and species occurrences.

Geologic outcrops, such as phonolites, increase microenvironmental heterogeneity, enhancing species richness at a landscape scale (Hjort et al., 2015). Increased speciation rates on isolated outcrops of scarce rocks are thought to lead to a higher

percentage of endemic species than the surrounding matrix (Ricketts, 2001). Geodiversity may thus promote both species richness and endemism. However, relatively little is known about the extent to which phonolites promote species diversity in general and particularly endemism. To approach this topic theoretically, phonolitic outcrops could be considered as small habitat islands within a basaltic matrix (Fig. 1b). The established 'species-area relationship' (SAR) and the 'species-isolation relationship' (SIR) (MacArthur & Wilson, 1967; Rosenzweig, 1995; Giladi et al., 2014) predict a smaller number of species on these small and isolated phonolitic rocks in comparison with basaltic rock outcrops in their surroundings. (From the beginning (MacArthur & Wilson, 1967), such concepts were not only meant for real islands but instead took 'islands' as examples for isolated habitats (or habitat islands) within a terrestrial landscape matrix.) However, the expected higher availability of nutrients would give such habitats more favourable conditions for plant growth. Besides, it is by no means certain that the phonolitic rocks were permanently separated from each other in southern La Palma's geological evolution. Possibly, a historically much larger phonolitic rock is today largely buried by basaltic eruptions (Garantje et al., 1998). Thus, a few phonolite outcrops may serve as refugia for remnant populations (Eriksson, 1996) of species specialised to phonolitic rocks.

We aim to investigate plant species richness and abundance on phonolites compared to surrounding basaltic lavas. To this end, we investigated the occurrences and traits of plant species in a comparative study matching basaltic and phonolitic rock formations on the island of La Palma.

La Palma hosts 159 vascular plant species that are endemic to the archipelago and 47 single island endemics (hereafter SIEs) (Beierkuhnlein et al. 2021). The endemic plant species *Cheirolophus junonianus* (Svent.) Holub, comprising its var. *junonianus* and var. *isoplexiphyllus* (Svent.) G.Kunkel (Vitales et al., 2014a; 2014b, Beierkuhnlein et al. 2021), occurs within a range of only 3500 m² solely on La Palma (Bañares et al., 2004). Within this small range, individuals of this species occur only on a few outcrops, which are almost exclusively phonolitic rocks with a chemical composition different from most of the surrounding substrates. Therefore, the species is very restricted in its range size to just a few small locations (Muer et al., 2016; Atlantis, 2021) and appears to be restricted to phonolites (Fig. 1a). This example species evokes the question if phonolites are of special importance for endemic species in the Canary Islands.

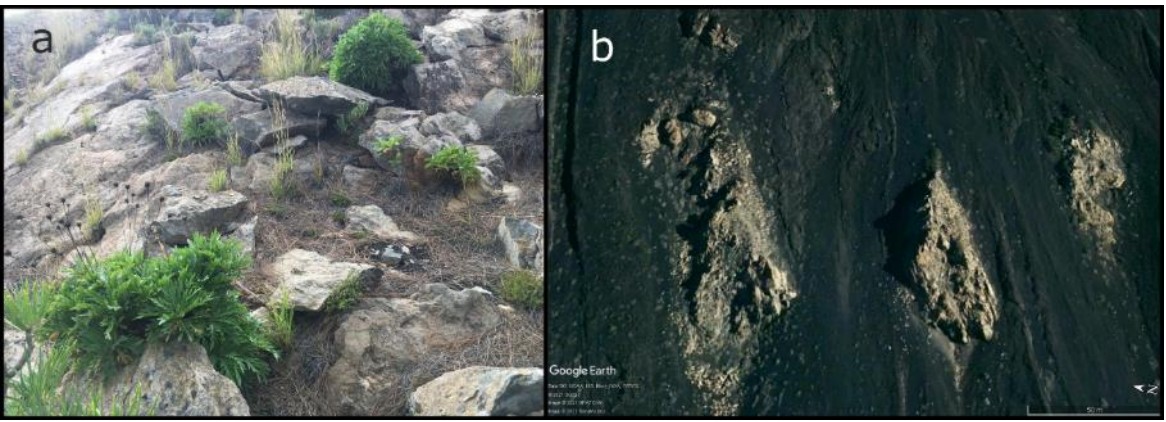

**Figure 1: The endemic *Cheirolophus junonianus* (a, bottom-left) and further plant species on a phonolite rock (© Severin Irl). Aerial image of rocks of phonolites isolated in a basaltic matrix in southern La Palma (b, © Google Earth 2020).**

We investigated the occurrences and traits of plant species in a comparative study matching basaltic and phonolitic rock formations on La Palma of comparable size, shape and extent to test the following hypotheses:

i. Species diversity: We expect plant species richness on phonolitic rocks to be higher than on basaltic rocks because phonolites offer more favourable plant growth conditions.

ii. Plant performance: Plant species populations on phonolites show a larger abundance of individuals that are taller and have greater canopy diameter than neighbouring basalts due to their advantages in resource availability and porosity. We used plant performance as a surrogate for plant fitness.

iii. Island endemism: We expect phonolitic rocks to host more endemic plant species than basaltic rocks because of their high degree of spatial isolation, in combination with the older age of the phonolitic bedrock than the basaltic matrix.

## 2 Methods

### 2.1 Study site and data sampling

We sampled four phonolitic and four adjacent basaltic rocks in the southern part of La Palma in spring 2018 (Walentowitz et al., 2021; Fig. 2). Locations were identified in the field based on Middlemost (1972). The sampled phonolitic rocks represented most of the overall extent of this habitat on the island, covering a large gradient of microclimate, aspect (northerness and easterness), and inclination. Local climate data are not available for individual plots, nor for the sites. Interpolated modelled

climate data (Karger et al., 2017) show only small variations in temperature and precipitation values for our study sites (Appendix A2). We chose comparable neighbouring pairs of phonolite and basalt consisting of one cohesive rock formation each. Outcrop pairs were chosen to match the size and microclimatic conditions (aspect, slope). For each selected phonolitic

and basaltic rock, we recorded plant species composition and abiotic parameters within 15 plots of 2 m x 2 m that were randomly selected within the range of accessibility on phonolite and basalt. This resulted in a total of 120 plots sampled across the four pairs of phonolitic and basaltic rocks (60 plots on phonolite and 60 plots on basalt).

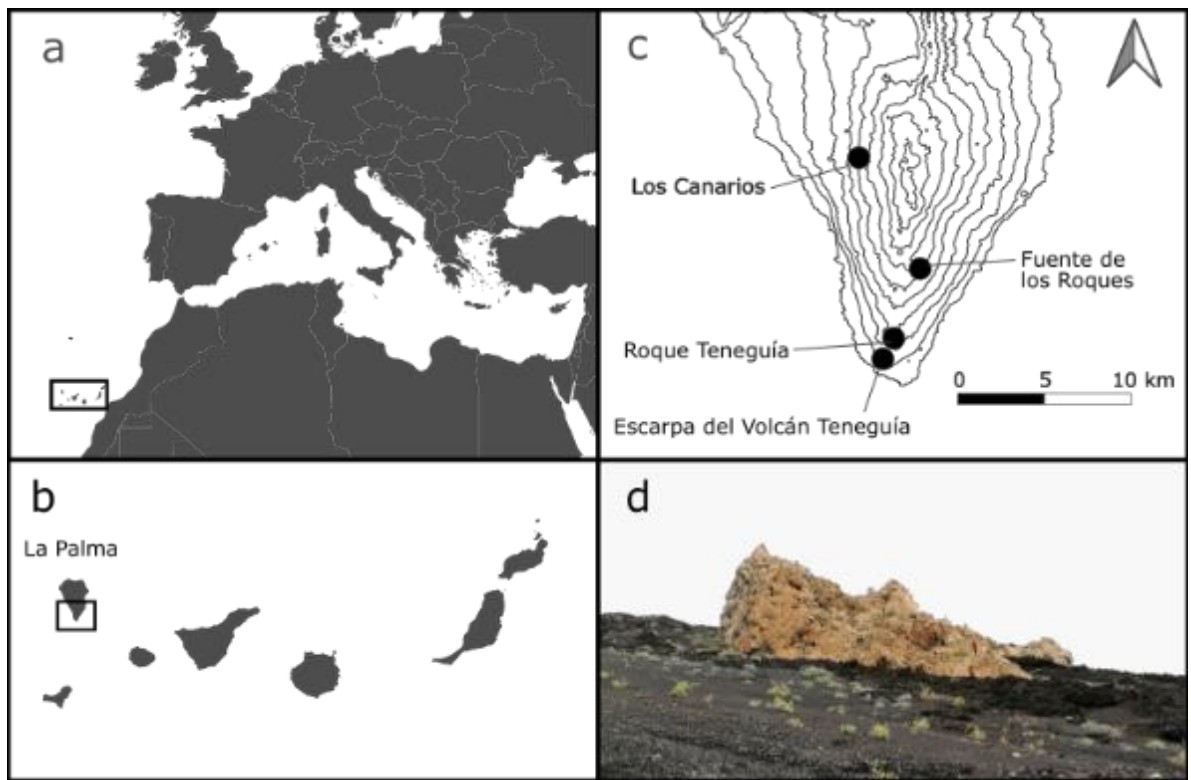

**Figure 2: Location of the Canary Islands (a) and La Palma (b). Southern La Palma with our 4 study sites and contour lines at 200**
**m intervals (c). Phonolite rock on Southern La Palma downwards slopes of the "Roque Teneguía", located in a basaltic matrix (d,**
**© Anna Walentowitz).**

Within each plot, we recorded coordinates, aspect, and slope inclination. Then, we estimated rock surface rugosity using thread transects spanning the two plot diagonals: we measured the transect length along the 3D-rock surface (Walentowitz et al.

2021). Larger values of rugosity indicate higher levels of microtopography (cracks, hollows, uneven slope), while low values indicate smooth, even surfaces.

All vascular plant species within each plot, including ferns, were identified following the taxonomy of Beierkuhnlein et al. (2021) that use "Plants of the World online" (POWO, 2019) as a taxonomic backbone. The biogeographic status of each species

(single-island endemic, multi-island endemic, non-endemic native and introduced) is based on Muer et al. (2016) (see extensive plant list in Appendix A4). The number of individuals per species and plot was counted.

Plant height (length from base of the stem to the tip) and canopy diameter (widest part of the plant parallel to the ground) of all single individuals found were measured as traits. Height, diameter, and species abundances were measured for all vascular
plant species. As plant communities were dominated by perennial species, we can expect that vegetational differences evolved through long-term processes and did not reflect the short-term variability of environmental conditions. We are aware that there is a serious debate on the trade-off between different functional traits and their effect on plant growth responses. However, we assume that height and diameter are good proxies for different components such as survival and reproduction that contribute to plant fitness (Laughlin et al., 2020). We furthermore know that numbers of flowers and seeds might be more accurate to
measure and monitore over the course of an entire reproductive cycle, but we chose plant height and width as proxies as these can be measured at the same time.

Lichen cover, which is abundant on the basalt, was estimated as the percent cover of each plot. Moss cover was negligible in all the plots.

**2.2 Statistical analysis**

Differences in total plant species number and the number of single and multi-island endemics were analysed using Pearson's Chi-squared tests. Percentages of abundance, plant height, diameter, and SIE percentage between plots on phonolites and basalt were analysed using Mann–Whitney U tests. We conducted detrended correspondence analysis (DCA) to investigate the multidimensional aspects of vegetation composition and identify potential fundamental underlying drivers (Appendix A5).
Afterwards, we applied a posthoc permutation test (10,000 repetitions) between the environmental variables (substrate, inclination, aspect and relief) and the DCA ordination axes (Appendix A6). We tested for differences in aspect, inclination, rock surface rugosity and lichen cover between phonolite and basalt using Mann–Whitney U tests.

**3 Results**

We recorded 68 species of vascular plants (pteridophtyes and spermatophtyes) overall. Of these species, nine were Single
Island Endemics (SIE) restricted to the island of La Palma, 16 were Multi Island Endemics (MIE) co-occurring also on other islands in the archipelago, 39 were non-endemic natives, and 4 were non-natives. The SIE *Cheirolophus junonianus* was only found on phonolite, and most individuals of var. *junonianus* occurred on one isolated outcrop (Roque Teneguía) and individuals of var. *isoplexiphyllus* on another one (Escarpa del Volcán Teneguía, Fig. 2c).

We found higher plant species richness on phonolitic rocks. While 22 species were encountered on both phonolite and basalt, only eleven species were restricted to basalt, but 34 were recorded only on phonolite (Appendix A3). Endemism groupings showed similar patterns (SIEs – phonolite: 9, basalt: 5; MIEs – phonolite: 15, basalt: 6). Besides the total number of plant species per rock type, we also found higher species richness on phonolite at the plot scale (p = 0.0164, Fig. 3 a), and higher diversity of SIEs (p = 0.00151, Fig. 3 b) and MIEs (p = 0.00727, Fig. 3 d). The percentage of SIEs (p = 0.1928, Fig. 3 c) and

MIEs (p = 0.05346, Fig 3 e) relative to total species number did not differ significantly at this scale between phonolitic and basaltic rocks.

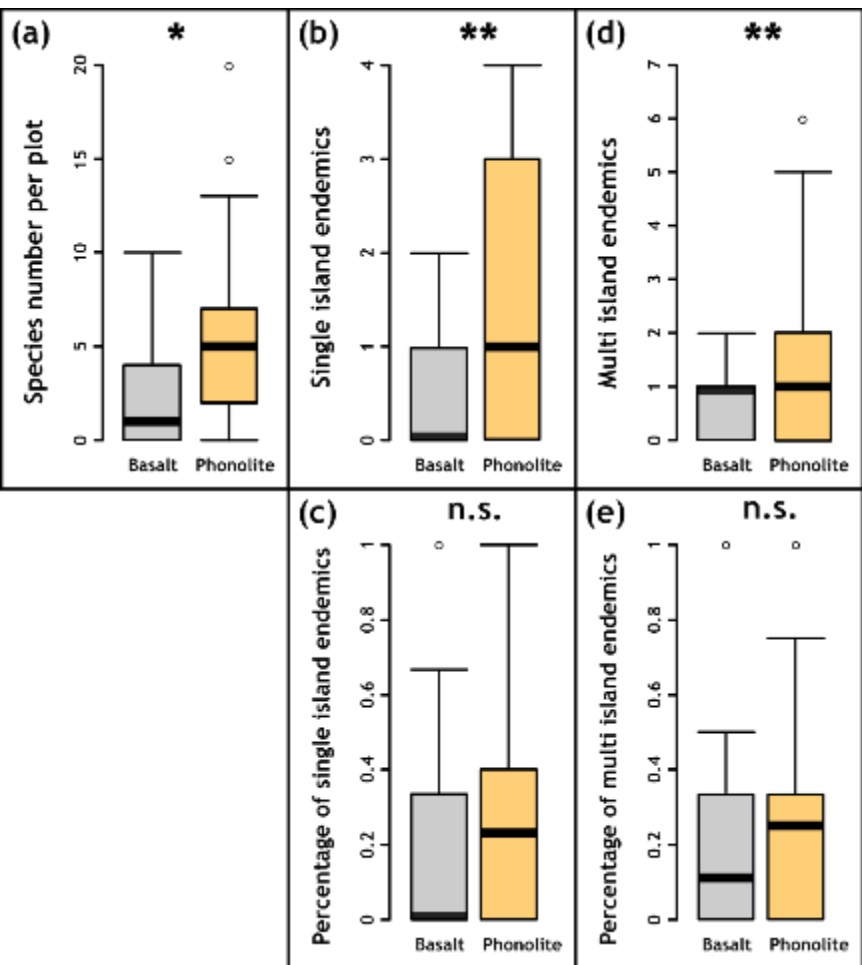

**Figure 3: Perennial species per 2 x 2 m plot for basaltic (n = 60) and phonolite substrates (n = 60). a) The number of species per phonolite plot is significantly larger than forbasaltic plots. b) Phonolites have significantly more single island endemics (SIE) and**

**(d) multi-island endemics. However, the numbers of endemic species relative to the total number of species do not differ significantly between substrates (c, e). All analyses were conducted with Pearson's Chi-squared test (a, b, c) and the Mann–Whitney U test (c, e).**

On phonolitic rocks, we did not find higher total plant abundance (p = 0.169, Fig. 4 a). Moreover, there was no significant difference in abundance when only considering the 23 species found in plots on both substrates (p = 0.179, Fig. 4 b).

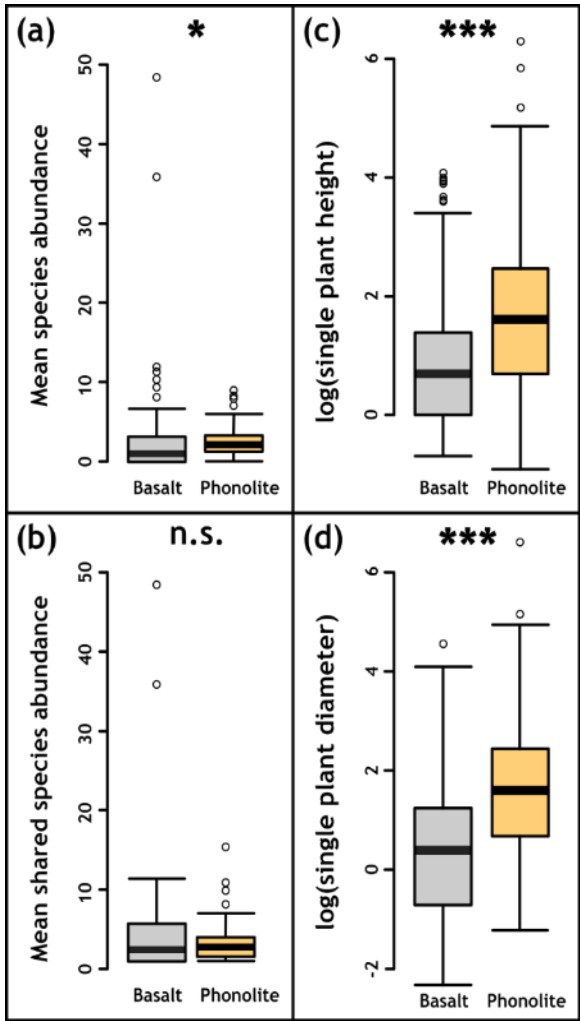

**Figure 4: a) Mean abundance differs significantly between basaltic (n = 60) and phonolite substrates (n = 60), but b) considering only shared species on both substrates resulted in no significant difference between basalt (n = 39) and phonolite (n = 51). c, d) Plant height and diameter (both log10-transformed) are significantly larger on phonolite plots (n = 1560) than on basaltic plots (n = 1173). Plot size: 2 x 2 m. All analyses were conducted with the Mann–Whitney U test.**

For plant species recorded on both rock types, individuals were on average taller and had wider canopies (Fig. 4c-d) on phonolitic than on basaltic rocks (Fig. 4c-d). Plant cover only (excluding lichens) was also significantly higher (p < 0.0001, Appendix A4a) on phonolites than on basalt. Lichen cover did not show a significant difference (p = 0548, Appendix A4b).

The ordination did not show any difference in the species composition, indicating no aspects of beta-diversity at all (Appendix A5 & A6). Topographic characteristics of basalt and phonolite plots showed no differences in surface rugosity, aspect (northerness, easterness), or differences in slope inclination (Appendix A7).

## 4 Discussion

The vegetation on phonolitic rocks differs compared to equivalent neighbouring basaltic rocks by by exhibiting higher species
richness and higher reproductive fitness of plant individuals and higher total plant cover. Higher numbers of single- and multi-island endemics on phonolites also reflect augmented total species numbers on this rock type.

Phonolite rocks on La Palma and other oceanic islands are arranged as habitat islands in a basaltic matrix (Fig. 1b). On La Palma, the total area of phonolite outcrops is tiny compared to the area of the basaltic matrix. Thus, encountering more species
on phonolites than on basalt aligns with our hypotheses but defies the area effect on species richness, which predicts species number to be lower in the phonolite habitat. Differences in species numbers might be accounted to lower-than-expected species growing on basalt or on higher-than-expected species numbers on phonolites. Our findings are congruent with studies, which did not find a species-area relationship or relationships with a less steep slope on habitat islands (Matthews et al., 2016; Deák et al., 2018). Reasons might be unrealised niches due to unsaturated evolutionary dynamics in this young and isolated system.
Environmental filters (Liu et al., 2020) enhancing growth conditions on phonolite outcrops may also exist, consistent with our findings that plants are larger on phonolites. Characteristics of rocks include, amongst others, texture, appearance, colour (potentially translating into differing rock surface temperature), fracture, streak and hardness. Consequently, mutually non-exclusive drivers may explain the observed phenomenon: 1) chemistry 2) age and 3) phonolite-specific surface texture and colour (as a proxy for rock surface temperature).


1) The rather porphyric arrangement of phonolites imply ameliorating plant growing conditions on this rock type. The chemical composition of phonolites has been described as nutrient-rich as it has traditionally been ground up and used as an inorganic fertilizer (von Wilpert & Lukes, 1998; Ramos et al., 2006; Schoen et al., 2016). Phonolites consist of the potassium-rich
nepheline, which dissolves much faster than other potassium sources (Manning, 2010). This may explain increased plant growth on phonolites. Weathering processes may act as driving forces for the establishment and germination of plants as they increase plant-available nutrients on these outcrops. Since ground phonolitic rock fertilizer is used to provide crops and trees with potassium (von Wilpert & Lukes, 1998; Schoen et al., 2016), plants growing on phonolites might profit from this rock characteristic. Basaltic rock powder has also been used as fertilizer in rare cases but according to Manning (2010) this is mainly
due to geologically unclear assignments (potassium-rich trachyte is often assigned to basalt). Other studies state that nutrient availability on basalt can be part of a feedback cycle where weathering releases essential nutrients that are key for plants to

survive but the occurrence of plants on basalt can also lead to higher availability of macronutrients (Ca, Mg, K), micronutrients (Fe), and non-essentials like Si and Na (Hinsinger et al., 2001). Although basalt and phonolite have both been reported to enhance nutrient conditions for plants, basalt was reported to be a potential but smaller source of potassium than phonolite (Manning 2010). Additionally, phonolite rocks in Southern La Palma are several hundred years older than the surrounding basalt (Carracedo et al. 1999) and have thus been exposed to weathering for longer timescales.

2) The age of geological formations influences plant diversity and species composition (Whittaker et al., 2008; Hulshof & Spasojevic, 2020). As noted in previous studies (Carracedo et al., 1999), the Cumbre Vieja rift on La Palma has evolved throughout several eruptions and therefore contains lava formations from different ages as well as slightly different mineralogical compositions. The current volcanic activity and lava flows at the Cumbre Vieja are a live example of this geological process (Pankhurst et al. 2021). The known phonolite rocks on La Palma are located in the geologically young southern part of the island. Consequently, we can assume that they are all in a stage of high ecological opportunities resulting in unsaturated niches (Whittaker et al. 2008). An interplay between erosion-driven uncovering of lava-covered phonolite rocks and new lava flows may strongly influence the vegetation on those rocks. We observed partly buried phonolites on which the survival of plants or seedlings during volcanic events was improbable (Garantje et al., 1998). Carracedo et al. (1999) showed that the last phonolite formation occurred in 1585, while there have been basaltic outbreaks until now (Pankhurst et al. 2021). Thus, age differences between phonolitic and basaltic outcrops might influence plant composition and performance. The younger age of basaltic rocks might have the consequence that evolutionary timescales were too short, niches remain empty, and basaltic outcrops host less species compared phonolites. In any case, the age of the substrate is significantly influencing the species numbers and plant performance. In addition, species composition might also be influenced by age differences of underlying rock material as 19 species including *Cheirolophus junonianus* can solely be encountered on phonolitic rocks (Muer et al., 2016). Interestingly, overall vegetation on phonolites cannot be considered to be distinct from basalt and these phonolite-specific species do not majorly drive vegetation patterns on this rock type. However, it is well known that habitat diversity on islands leads to higher species richness (Hortal et al., 2009). Phonolites offer an additional habitat for plants to grow on and hereby contribute to plant species richness on La Palma (Irl et al., 2015).

3) Besides petro-chemical characteristics and rock age, the surface structure and colour of phonolites might be suspected to drive plant patterns on such rocks. We observed deeper fractures in phonolitic rocks than in other volcanites on La Palma. Besides, phonolitic rocks show a much smoother surface roughness than their surrounding matrix. Basaltic rocks seem to possess a more dynamic relief, mainly attributed to their origin in congealed lava flows, typically found on oceanic islands. Nevertheless, when testing rock surface rugosity, there were no significant differences between phonolitic and basaltic rocks. Hence, we argue that surface characteristics do not play a role for higher plant growth response, richness and abundance observed on phonolites. Besides fractures, another visual observation was that phonolites are of lighter colour than their

surrounding basaltic matrix. We expected that phonolites hold a higher albedo than surrounding rocks and therefore expected them to have a reduced surface temperature compared to volcanic outcrops with darker colouring, such as basalt. However, in an experiment with differently coloured bricks Hall et al. (2005) showed that the albedo of white surfaces only leads to significantly lower temperature of the material when the surface temperature falls below air temperature. With monthly temperatures between 17 and 25° C within large areas of oceanic islands (Harter et al., 2015), no major temperature differences between basalt and phonolite surfaces can be expected. We therefore consider that this effect has no major impact on plants' habitat suitability.

While rock chemistry and age seem to increase species numbers on phonolites, these rock characteristics do not lead to higher percentages of endemic plants or a compositional distinct vegetation. Even though the endemic *Cheirolophus junonianus* with its two varieties var. *junonianus* and var. *isoplexiphyllus* is confined to phonolites, the percentage of single island endemics on phonolites was not significantly higher, refuting our expectations. As most individuals of the typical variety of *Cheirolophus junonianus* occur on one isolated outcrop and individuals of var. *isoplexiphyllus* on another one (personal observation), it could be suspected that a very local allopatric speciation by adaptive radiation or an ongoing genetic drift could be the underlying cause (Vitales et al., 2014a; 2014b). Thus, the differential geology of phonolites itself does not result in a specialized flora of habitat islands, contrary to e.g., Kruckerberg (1991), and populations of *Cheirolophus junonianus* must be seen in another context. Considering the geologic history of the islands South (Garantje et al., 1998) it is a more  possible explanation that this singularity presumes that *Cheirolophus junonianus* belongs to a relict population of plants that were once widely distributed on phonolitic rock before these were covered mainly by basalt. Consequently, lessons learned from other outcrops (Kruckerberg, 1991) cannot be adapted to the phonolitic rocks on La Palma, and the functioning of phonolites as islands of speciation within a matrix of basalt does not seem to apply. However, considering *Cheirolophus junonianis* to be a remnant plant population on phonolites raises, on the one hand, the question if the species might increase community resilience and ecosystems stability, including nutrient cycling, as hypothesized by Eriksson (2000) for remnant populations in general. On the other hand, possible extinction debts might lead to the disappearance of the species in the future and only the continuous monitoring of these populations can help to answer these questions.

When looking at the role of geologic promotion of biodiversity, there are several studies addressing serpentinite rocks and serpentine soils. On those rocks and soils, low amounts of essential nutrients (N, K, and P), and a low calcium-to-magnesium ratio relate to high rates of endemism and a specialized flora (Chiarucci, 2003; Harrison et al., 2006). These serpentinite-tolerating species are restricted to a harsh environment by dominant competitors in a less harsh matrix. On phonolitic rocks, more favourable growing conditions resulting in higher plant richness and abundances seem to prevail. Compared to serpentinite rock studies, the surrounding matrix is built up by the potentially harsher basaltic rocks resulting in lower plant growth response.

The geological history of basaltic rocks depends on a series of different volcanic eruptions, assuming a more considerable chemical variability, whereas phonolites seem to build upon one or just a few volcanic events (Carracedo et al., 1999). The long chronology of volcanic eruptions on the Canary Islands and their ancestors reveals a high likeliness that most of the eruptions resulted in basaltic, only a few in phonolitic formations. Plants growing on basalt experienced larger environmental gradients since basalt is omnipresent on the islands. Contrasting, plants growing only on phonolites did not experience larger

environmental gradients.

We are not aware of other studies conducted in locations where phonolites can be encountered that explore their potential role as exceptional plant habitat islands, even though phonolites can be found all over the world (Garcia et al., 1986; Ackerman et al., 2015; Hagos et al., 2017). Therefore, further investigation is needed to investigate whether the patterns encountered on La

Palma may also be found on comparable phonolitic rocks in other areas of the world. Their benefits for biodiversity found in this study need to be recognized and valued. Especially for isolated areas such as islands, phonolites can contribute to small-scale biodiversity hotspots and our findings suggest that they should be conserved.

## 5 Conclusion

Phonolites provide unique habitat conditions for plants on oceanic islands compared with surrounding areas. Higher species

numbers and abundances as well as higher plant performance underline the importance of these rocks for the vegetation on oceanic islands. Despite the small total area covered by phonolites they play a significant role in enhancing plant biodiversity on the island of La Palma. Our results suggest that exceptional rock outcrops like phonolites contribute to a better understanding of the formation of plant diversity on volcanic islands. As oceanic islands have always been formed through volcanic activity on the oceanic crusts, the combination of basaltic and phonolitic rocks is highly likely a regular pattern in

Earth history.

## Appendix

**A1: References of selected chemical components of basaltic and phonolitic substrates. B) indicates basaltic substrats or treatments, P) phonolitic and N) nepheline syenite (phonolite equivalent with larger grain size) ones.**

| Study | System/ study | Ca | Mg | K | Mn | P | Fe |
|-------|---------------|-----|-----|-----|-----|-----|-----|
| Roqueto do Reis (2021; thesis) | Substrate used in experiments | B) 8.54% P) 1.76% CaO | B) 4.74% P) 0.32% MgO | B) 1.25% P) 8.05% $K_2O$ | B) 0.21% P) 0.25% MnO | B) 0.42% P) 0.07% $P_2O_5$ | B) 14.82% P) 3.87% $Fe_2O_3$ |

| Garcia et al. (1986) | Field work (Kaula Isl.) | P) 1.74% CaO | P) 1.93% MgO | P) 4.48% K$_2$O | P) 0.31% MnO | P) 0.64% P$_2$O$_5$ | P) 3.33% Fe$_2$O$_3$; P) 1.74% FeO |
|---|---|---|---|---|---|---|---|
| Hagos et al. (2017) | Field work (Axum) | P) 1.12% CaO | P) 0.05% MgO | P) 4.94% K$_2$O | P) 0.30% MnO | P) 0.04% P$_2$O$_5$ | P) 5.47% Fe$_2$O$_3$ |
| Manning (2010) | Review | B) 9.47% N) 2.31% CaO | B) 6.73% N) 0.77% MgO | B) 1.10% N) 5.58% K$_2$O | B) 0.20% N) 0.15% MnO | B) 0.35% N) 0.13% P$_2$O$_5$ | B) 3.79% N) 2.25%; Fe$_2$O$_3$ B) 7.13% N) 2.05% FeO |

**A2: Interpolated data from Climate models for our research sites bases on the Chelsa Climate Data (Karger et al., 2017).**

| Site | Annual mean temperature [C °] | Annual precipitation [mm] |
|---|---|---|
| Los Canarios | 16.7 | 651 |
| Fuente de los Roques | 18.2 | 536 |
| Roque Teneguía | 16.5 | 633 |
| Ecarpa del Volcán Teneguía | 18.8 | 517 |

**A3: Complete list of all study species encountered on phonolites (P) and basalt (B) including their status as single island endemic (SIE), multi-island endemic (MIE), native (nat.) and introduced (intr.). The taxonomy follows the standards of "Plants of the World online" (POWO 2019) updated and adapted to the FloCan Checklist (Beierkuhnlein et al., 2021).**


| Species | Family | Rock type | Status | woody | perennial |
|---|---|---|---|---|---|
| *Aeonium arboreum* ssp. *holochrysum* (H.Y.Liu) Bañares | Crassulaceae | B/P | MIE | 1 | 1 |
| *Aeonium davidbramwellii* H.Y.Liu | Crassulaceae | B/P | SIE | 1 | 1 |
| *Aeonium diplocyclum* (Webb ex Bolle) T.H.M.Mes | Crassulaceae | B | MIE | 1 | 1 |
| *Aichryson bollei* Webb ex Bolle | Crassulaceae | P | SIE | 0 | 1 |
| *Aira caryophyllea* L. | Poaceae | B/P | nat. | 0 | 0 |
| *Allium canariense* (Regel) N.Friesen & P.Schönfelder | Amaryllidaceae | P | MIE | 0 | 1 |
| *Anogramma leptophylla* (L.) Link | Pteridaceae | B | nat. | 0 | 0 |
| *Anthoxanthum odoratum* L. | Poaceae | P | nat. | 0 | 1 |
| *Arabidopsis thaliana* (L.) Heynh. | Brassicaceae | B/P | nat. | 0 | 0 |
| *Arenaria leptocladus* (Rchb.) Guss. | Caryophyllaceae | B/P | nat. | 0 | 0 |
| *Argyranthemum haouarytheum* Humphries & Bramwell | Asteraceae | P | SIE | 1 | 1 |
| *Astydamia latifolia* (L.f.) Baill. | Apiaceae | P | nat. | 1 | 1 |
| *Bituminaria bituminosa* (L.) C.H.Stirt. | Fabaceae | P | nat. | 1 | 1 |

| | | | | | |
|---|---|---|---|---|---|
| *Brassica oleracea* L. | Brassicaceae | B | intr. | 0 | 1 |
| *Bystropogon origanifolius* var. *palmensis* | Lamiaceae | B/P | SIE | 1 | 1 |
| *Cardamine hirsuta* L. | Brassicaceae | B | nat. | 0 | 0 |
| *Cheirolophus junonianus* (Svent.) Holub | Asteraceae | P | SIE | 1 | 1 |
| *Cosentinia vellea* ssp. *bivalens* (Reichstein) Rivas Mart. & Salvo | Pteridaceae | B/P | nat. | 0 | 1 |
| *Crassula campestris* (Eckl. & Zeyh.) Endl. | Crassulaceae | B | intr. | 0 | 0 |
| *Davallia canariensis* (L.) Sm. | Davalliaceae | B/P | nat. | 0 | 1 |
| *Echium brevirame* Sprague & Hutch | Boraginaceae | B/P | SIE | 1 | 1 |
| *Erica arborea* L. | Ericaceae | P | nat. | 1 | 1 |
| *Erigeron bonariensis* L. | Asteraceae | P | nat. | 0 | 0 |
| *Erodium botrys* (Cav.) Bertol. | Geranicaeae | P | nat. | 0 | 0 |
| *Festuca muralis* Kunth | Poaceae | B | nat. | 0 | 0 |
| *Filago germanica* (L.) Huds. | Asteraceae | B | nat. | 0 | 0 |
| *Galium aparine* L. | Rubiaceae | B/P | nat. | 0 | 0 |
| *Geranium molle* L. | Geraniaceae | P | nat. | 0 | 0 |
| *Geranium purpureum* Vill. | Geranicaeae | P | nat. | 0 | 0 |
| *Hemionitis gluckuk* Christenh. | Pteridaceae | P | nat. | 0 | 1 |
| *Hemionitis guanchica* (Bolle) Christenh. | Pteridaceae | B/P | nat. | 0 | 1 |
| *Holcus lanatus* L. | Poaceae | P | nat. | 0 | 1 |
| *Hyparrhenia hirta* (L.) Stapf | Poaceae | B/P | nat. | 0 | 1 |
| *Kleinia neriifolia* Haw. | Asteraceae | P | MIE | 1 | 1 |
| *Lavandula canariensis* Mill. | Lamiaceae | P | MIE | 1 | 1 |
| *Lobularia canariensis* (DC.) L.Borgen | Brassicaceae | P | MIE | 1 | 1 |
| *Medicago truncatula* Gaertn. | Fabaceae | P | nat. | 0 | 0 |
| *Mercurialis canariensis* Obbard & S.A.Harris | Euphorbiaceae | P | MIE | 0 | 0 |
| *Micromeria herpyllomorpha* Webb & Berthel. | Lamiaceae | B/P | SIE | 1 | 1 |
| *Monanthes muralis* (Webb ex Bolle) Hook.f. | Crassulaceae | B/P | MIE | 0 | 1 |
| *Ononis serrata* Forssk. | Fabacea | P | nat. | 0 | 0 |
| *Opuntia ficus-indica* (L.) Mill. | Cactaceae | P | intr. | 1 | 1 |
| *Parietaria debilis* G.Forst. | Urticaceae | P | nat. | 0 | 0 |
| *Paronychia canariensis* (L.f.) Link | Caryophyllaceae | P | MIE | 1 | 1 |
| *Periploca laevigata* Aiton | Apocynaceae | P | nat. | 1 | 1 |
| *Phagnalon purpurascens* Sch.Bip. | Asteraceae | P | nat. | 1 | 1 |
| *Pinus canariensis* C.Sm. ex DC. | Pinaceae | P | MIE | 1 | 1 |
| *Polycarpaea aristata* (Aiton) C.Sm. ex DC. | Caryophyllaceae | B/P | MIE | 0 | 1 |
| *Polycarpaea tenuis* Webb ex Christ | Caryophyllacea | P | MIE | 0/1 | 1 |
| *Polypodium macaronesicum* A.E.Bobrov | Polypodiaceae | B/P | nat. | 0 | 1 |

| | | | | | |
|---|---|---|---|---|---|
| *Pteridum aquilinium* (L.) Kuhn | Pteridaceae | P | nat. | 0 | 1 |
| *Pterocephalus porphyranthus* Svent. | Caprifoliaceae | P | SIE | 1 | 1 |
| *Rubia fruticosa* Aiton | Rubiaceae | P | nat. | 1 | 1 |
| *Rumex bucephalophorus ssp. canariensis* (Steinh.) Rchb.f. | Polygonaceae | B | nat. | 1 | 1 |
| *Rumex lunaria* L. | Polygonaceae | B/P | MIE | 1 | 1 |
| *Schizogyne sericea* (L.f.) DC. | Asteraceae | B/P | nat. | 1 | 1 |
| *Sideritis barbellata* Mend.-Heuer | Lamiaceae | B/P | SIE | 1 | 1 |
| *Solanum villosum* Mill. | Solanaceae | P | nat. | 0 | 1 |
| *Sonchus hierrensis* (Pit.) Boulos | Asteraceae | P | MIE | 1 | 1 |
| *Sonchus oleraceus* L. | Asteraceae | B/P | nat. | 0 | 0 |
| *Stachys arvensis* (L.) L. | Lamiaceae | B/P | nat. | 0 | 0 |
| *Todaroa aurea* (Aiton) Parl. | Apiaceae | P | MIE | 0 | 1 |
| *Tolpis laciniata* Webb | Asteraceae | B/P | MIE | 0 | 1 |
| *Trifolium arvense* L. | Fabaceae | B | nat. | 0 | 0 |
| *Tuberaria guttata* (L.) Fourr. | Cistaceae | B | nat. | 0 | 1 |
| *Umbilicus gaditanus* Boiss. | Crassulaceae | B/P | nat. | 0 | 1 |
| *Valeriana dentata* (L.) All. | Valerianaceae | P | intr. | 0 | 0 |
| *Wahlenbergia lobelioides* (L.f.) Link ssp. *lobelioides* | Campanulaceae | B | nat. | 0 | 0 |

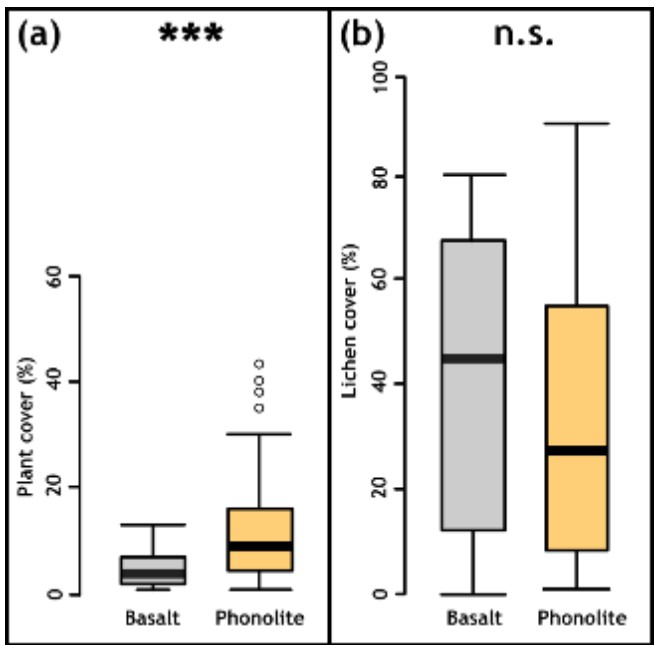

A4: Vegetation cover on basaltic and phonolite plots. (a) Plant cover showed significant (p < 0.001, Mann–Whitney U test) and (b) lichen cover showed no significant difference between the substrates (p > 0.05, Mann–Whitney U test).


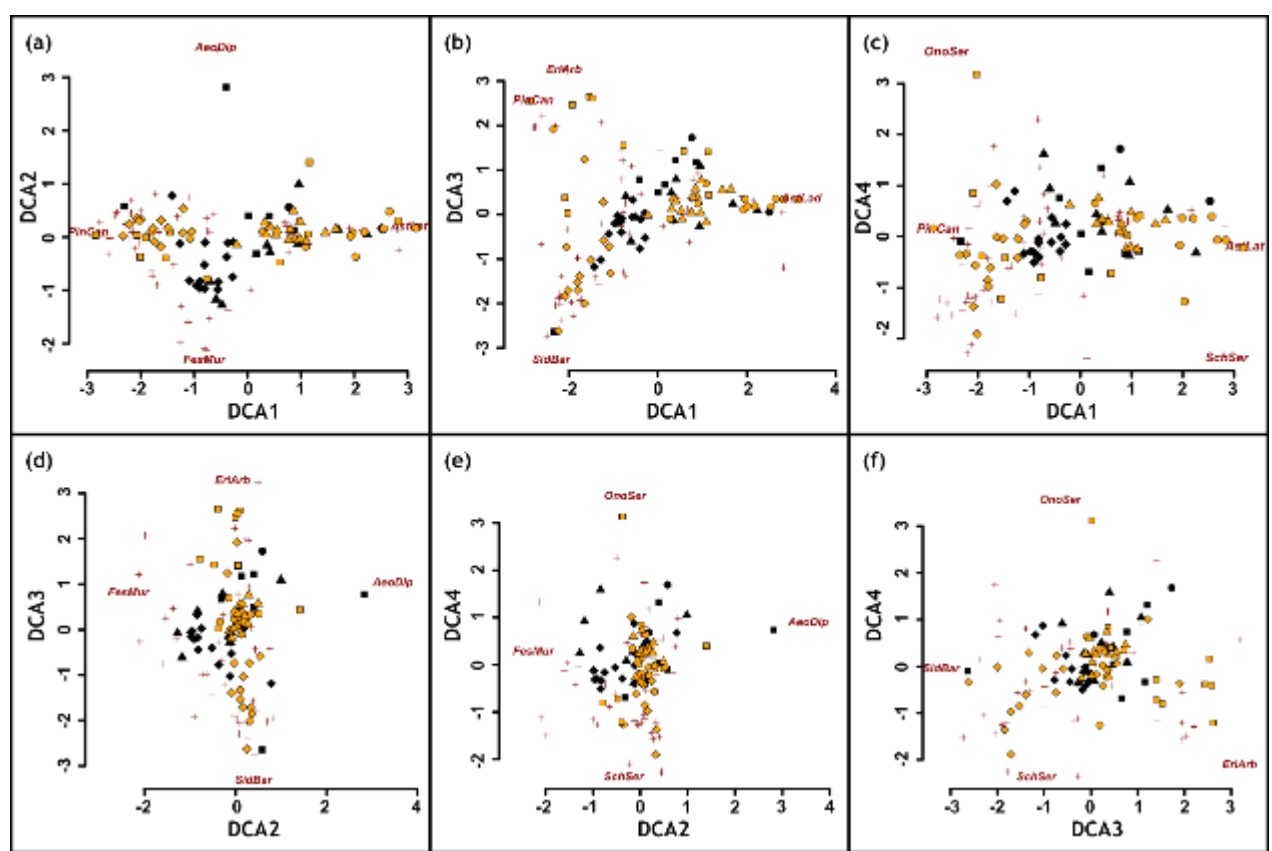

**A5: Detrended Correspondence Analyses (DCA) shows no clear difference between phonolite and basaltic rock vegetation. Yellow dots show phonolite plots, black dots basalt plots, dark red a subset of species centroids. Species names chosen based on most extreme values along the gradients.**

**A6: A posthoc permutation test (10,000 repetitions) between the DCA ordination axes and the environmental variables (substrate, inclination, aspect and relief) showed no significant differences between phonolite and basalt. Obviously, the variation shown in the DCA does not depend on the substrate (but there is a relationship between northerness and the fourth dimension DCA4).**

|  | DCA1&DCA2 | DCA1&DCA3 | DCA1&DCA4 | DCA2&DCA3 | DCA2&DCA4 | DCA3&DCA4 |
|---|---|---|---|---|---|---|
| **Substrate** | p = 0.623 | p = 0.503 | p = 0.768 | p = 0.289 | p = 0.959 | p = 0.439 |
| **Inclination** | p = 0.490 | p = 0.946 | p = 0.315 | p = 0.523 | p = 0.108 | p = 0.365 |
| **Northerness** | p = 0.914 | p = 0.526 | p = 0.032 | p = 0.875 | p = 0.933 | p = 0.921 |
| **Easterness** | p = 0.293 | p = 0.564 | p = 0.310 | p = 0.429 | p = 0.213 | p = 0.426 |
| **Rugosity** | p = 0.212 | p = 0.387 | p = 0.324 | p = 0.875 | p = 0.933 | p = 0.921 |

**A7: Environmental plot characteristics. Inclination on phonolites was (despite efforts to sample similar environments) significantly higher than on basalt. Components of exposition (northerness and easterness) and rugosity showed no significant differences (unpaired Whitney test).**

|  | Rugosity (m) | Northerness | Easterness | Inclination (°) |
|---|---|---|---|---|
| **Mean Basalt** | 3.557 | 0.01407 | 0.04970 | 43.0 |
| **Mean Phonolite** | 3.643 | -0.09062 | 0.04303 | 53.8 |
| **p-Value** | 0.7781 | 0.6525 | 0.8827 | 0.0277 |

## Code availability

Only standard tests and plotting commands in R were used for data analysis. The code is available on request from the corresponding author.

## Data availability

Any data supporting the findings of this study are available within the supplementary materials of this article and were taken from Walentowitz et al. (2021).

## Authors contributions


C.B., S.D.H.I., D.K., L.S. and A.J.W. developed the research idea, D.K., L.S. and A.J.W. conducted the field work, analysed the data and led the writing process. All authors developed the methods, discussed the results and contributed to the manuscript.

## Competing interests

The authors declare that they have no conflict of interest.

## Acknowledgements


We would like to kindly thank the technical staff of the Biogeography Department of the University of Bayreuth for supporting this study and being of immense help in the implementation of field work. The Caldera de Taburiente National Park Directorate and especially Felix Medina from the Consejería de Medio Ambiente, Cabildo de La Palma, are being thanked for permitting investigations in protected areas on La Palma and for their expertise on the local flora. This project has received funding from

the European Union's Horizon 2020 research and innovation programme ECOPOTENTIAL under grant agreement No 641762. The work on this study is additionally supported by the pilot mySpace in the Horizon 2020 project e-shape under grant agreement No 820852.

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
