# Peer review of "Geodiversity and biodiversity on a volcanic island: The role of scattered phonolites for plant diversity and performance"

_Biogeosciences, 2021_

## Author Response (AR1)

The authoring team would like to thank the reviewers for the constructive and thoughtful review of the manuscript. On the basis of the reviewer's comments, we edited and complemented the manuscript. The discussion and the conclusion received most changes. Below we addressed all comments of both reviewers in detail.

**Anonymous Referee #1, 15 Jun 2021**

**General comments**

The authors present an experimental study conducted on the island of La Palma, where they studied the effect of different substrates in the form of pholotitic and basaltic rocks on plant performance and species abundance. I think the study is relevant, the experiment well set-up and the conclusions interesting. I do have suggestions to potentially improve the paper, particularly regarding the framing and discussion of the results presented in the paper.

The first point of attention is that the overall conclusion of the paper, formulated as 'Phonolites host distinct vegetation compared to equivalent areas of neighbouring basaltic rocks', is not supported by the data, in particular the Detrended Correspondence Analyses presented in Figure A3, which shows 'no clear difference between phonolite and basaltic rock vegetations'. I think the conclusions of the paper should be reformulated to better reflect the results of the study.

**Reply:** Thank you very much for this comment concerning the conclusion. We agree that it is not the distinct vegetation that make phonolites special, but higher species numbers and plant performance. The term "distinct vegetation" was removed from the conclusion. The focus is now purely on higher species numbers and plant performance on phonolites compared to volcanic outcrops of basaltic origin.

A second point of attention is the lack of detail provided in the differences in the petrographic and geochemical differences of the phonolites and basaltic substrates. I think that in particular, the availability of nutrients in these two substrates should be reported, as this information is key to the interpretation of the results discussed in 3) (L.243-253). It would be preferable to conduct and report measurements of the nutrient contents in the plots, but if not possible, I would at least expect a more detailed description of the chemical differences in nutrient availability reported in literature. Given the results of this study and it's potential to reach a more biologically minded audience, I also think it is worth elaborating how phonolites differ from the basaltic substrates in their chemical composition (in L.74 for example), which is mentioned, but not elaborated on. Also, the presence of heavy metals is discussed in L.281-287 to explain the differences that are reported in the paper, which calls for these to be measured as well.

**Reply:** Thank you very much for this remark on differences in chemical characteristics of phonolites and basalt. We followed your advice and included information from the literature on nutrient availability and chemical composition of both rock types in the discussion (L. 299ff. in MS with track-changes). We anticipate that we now have a more detailed description of the chemical differences in nutrient availability, allowing a profound basis for discussion. Measurement of nutrient availability in the field was not possible under the given sampling conditions, and therefore studies like Schoen et al. (2016) and Manning (2010) serve as sources on phonolites' chemical composition. Furthermore, we included literature to allow comparison of nutrient release from basalt and phonolite rock powder (e.g., Hinsinger et al. 2001, Tavares et al. 2018, Nogueira et al. 2021).

Statements concerning the presence of heavy metals were removed from the manuscript as we cannot provide measured petro-chemical evidence that supports what can be found in the literature.

I think the introduction can be streamlined. Currently, the introduction works towards the knowledge gap that is introduced in L.109. However, in between the introduction and the knowledge gap there is a considerable amount of text that introduces the study site of La Palma, which in my opinion, breaks the flow of the introduction. I would suggest to bring the paragraph where the knowledge gap is formulated forward, then following with a statement along the lines of: 'To this end, we investigated the occurrences and traits of plant species in a comparative study matching basaltic and phonolitic rock formations on the island of La Palma.' Followed by an introduction of the study site and its relevance to this study. Also, I think the paragraph in L.113-122 lead to the knowledge gap and should be integrated in the previous paragraph, and should be directly followed by the knowledge gap.

**Reply:** Thank you very much for these suggestions on improving the introduction. Accordingly, we moved the paragraph concerning the knowledge gap (L. 109 ff.) forward (with slight changes) and put paragraph lines 113-22 directly above this part of the text. Additionally, paragraph lines 113-22 were integrated into the previous paragraph to streamline the introduction.

I object to how the second hypothesis is currently formulated. The hypothesis is currently introduced with species *reproductive* fitness, which is then stated to be proxied with plant height and canopy diameter. I fully understand the choice to measure these two metrics due to the time constraints of a field work abroad, I don't think these can be used as a proxy for *reproductive* fitness. Instead, I suggest presenting these as proxies for plant performance, or maybe as proxies for plant fitness (not reproductive fitness), stating that these metrics are good proxies for the different components that make up plant fitness (e.g. survival and reproduction, see Laughlin 2020). I would also introduce the use of plant height and canopy diameter as proxies for plant performance/fitness in the methods section.

Laughlin, D. C., J. R. Gremer, P. B. Adler, R. M. Mitchell, and M. M. Moore. 2020. The net effect of functional traits on fitness. Trends in Ecology & Evolution.

**Reply:** We appreciate this advice very much, especially the hint to the paper from Laughlin et al. (2020). We agree that the term plant performance fits better to our study. The authoring team had several discussions on the terms reproductive fitness, fitness, etc. before submission and again after receiving this comment. We will follow your suggestion to use 'plant performance' but give some hints that we want to address 'plant fitness'. Indeed, the trade-offs between growth, survival and reproduction make it more sophisticated to use traits as proxies to quantify the effects of fitness. We rephrased the second hypothesis as following (L. 169 in MS with track-changes):

"Plant performance: Plant species populations on phonolites show a larger abundance of individuals that are taller and have greater canopy diameter than neighbouring basalts due to their advantages in resource availability and porosity. We used plant performance as a surrogate for plant fitness."

Furthermore, we added the details to the question of why we measured plant performance using the proxies plant height and canopy diameter with reference to Laughlin et al. (2020) to the methods section (L. 205-2014 in MS with track-changes).

In addition, we also replaced the word "fitness" with "performance" in the title of the manuscript.

I fail to see the relevance of the specific introduction and discussion of the SIE Cheirolophus junonianus to the story of this paper. Appendix A1 shows that more species occur on only basalts or phonolites, but this is only discussed in the context of this one species. I would expect this to lead to differences in the species composition of the different substrates, yet the data presented in appendix A3 shows no compositional differences between substrates. This is highly relevant to the paper, and I would like to see this discussed more, and maybe also analysed in more detail. In the paper, the authors also mention that the basaltic rocks house more generalists, but no analysis is presented that shows the what species are considered specialists or generalists.

**Reply**: Thank you for this substantial and detailed comment. We rephrased paragraphs about *Cheireolophus junonianus* to make it more understandable why this species was chosen to be part of the storyline (L. 131ff, 334ff in MS with track changes). The species is emphasized in the paper, as it is known to almost exclusively grow on phonolites. We kept this species as it initially stimulated the development of hypotheses, and it adds an ecosystem perspective to our paper, but at the same time clarify that one species does not lead to significantly distinct vegetation on phonolites and basalt and instead seems to be the exception than the rule. Furthermore, we added a paper by Eriksson (2000) to show why remnant populations of plants, such as *Cheirolophus junonianus* on phonolites, can be of importance for community stability and even influence nutrient cycling.

We furthermore did some more analyses on the compositional data between the substrates. Interestingly, a posthoc permutation test (10,000 repetitions) between the DCA ordination axes and the environmental variables (substrate, inclination, aspect, relief) showed no significant differences between phonolite and basalt (p-values: DCA2~DCA1: 0.623; DCA3~DCA1: 0.503; DCA4~DCA1: 0.768; DCA3~DCA2: 0.289; DCA4~DCA2: 0.959; DCA3~DCA4: 0.439). Apparently, the variation shown in the DCA does not depend on the substrate (but there is a relationship between northerness and the fourth dimension DCA4).

We agree that the data presented it not supporting any assumptions concerning generalists vs. specialists. We, therefore, deleted this part from the manuscript to stick to the assumptions we can draw from our data.

I would suggest restructuring the discussion The wording in L.226 suggests that these four drivers are expected to have an effect, so I was disappointed to then read the conclusions on the first two paragraphs (1. and 2.), telling me that these aspects didn't play a role. While not irrelevant, I would start with the points 3 and 4, which are more relevant to the discussion, and then bundle points 1 and 2 into a paragraph highlighting some other drivers that were not expected to play a role in this system.

**Reply**: Yes, we agree that changing the order of points 1-4 increases the quality of the discussion. The discussion now starts with the more relevant aspects, which in its new order are 1) chemistry and 2) age. In point 3 we discuss rock surface structure and colour in one paragraph as suggested.

I think the point discussed in 4) is very relevant and interesting, and I see an opportunity to place this research into a broader context that the authors didn't explore in full. I think the result on species abundances can be caused by either a lower than expected species richness on the basalts, or a higher than expected species abundance on phonolites, or both. The age difference between these two substrates might suggest that there are unfilled niches in on the basalt substrates (which is mentioned,

but I think can be elaborated on), but also that there might be an extinction deficit in the phonolite habitats due to habitat decrease and habitat fragmentation. I think this opens up the possibility to discuss how geological history affects evolutionary processes. In line with that, I think the discussion can come back to the relevance of this work to understand how geological diversity affect biodiversity, as introduced in the opening paragraphs of the introduction.

**Reply**: Thank you very much for this inspirational comment that adds tremendous value to the discussion of the paper. Due to the importance of this point, we now mentioned age differences between substrates potentially affecting vegetation earlier on in the discussion (now point 2). Additionally, we added aspects discussing the possibility that it might not be only phonolites exhibiting higher species numbers but that it might also be that basalt hosts lower species numbers due to age.

The aspect of phonolites hosting vegetation with potential extinction debt has been included in L. 376f (in MS with track-changes).

Closing the circle in the discussion and coming back to the initial questions of how geodiversity affects biodiversity leads to a slightly changed order of paragraphs in the manuscript.

**Detailed comments**

I also have some specific and technical comments:

L.40: chemical composition

**Reply**: Thank you. We changed 'geological elements (i.e., composition)' to 'chemical components'.

L.55: Omit 'also'

**Reply**: We rephrased the sentence to avoid ''also'.

L.56-59: I think this sentence needs to be rewritten to better connect to the previous sentence and align with the rest of the paragraph. Perhaps change the other of the sentence? 'This underlines the relevance of understanding the importance of geodiversity for insular biodiversity, which is particularly vulnerable to extinction due to restricted ranges and small population sizes of insular endemic species (Paulay, 1994).'

**Reply**: Thank you for this excellent idea to improve the connection between the two sentences. We followed your advice and kept 'highly restricted ranges'.

L.65: 'Distinctive to phonolites is fine-to-medium grain size'- Is there any relevance of this characteristic to this study?

**Reply**: Thank you for this comment. We decided to delete this sentence.

L.97: What does hyper-endemic means in this context? I suspect the authors mean this is a single-island endemic, and I would refer to it as such. (Note that hyper-endemic is a term used in epidemiology, referring to persistent, high levels of disease occurrence).

**Reply**: Thank you for bringing this up. We wanted to underline with "hyper-endemic" that the species occurs only on two rock samples on La Palma. Nevertheless, in the same paragraph, we explain that the species "occurs within a range of only 3500 m²" so we decided to delete "hyper" to avoid being misunderstood since the term is commonly used in epidemiology.

L.153: Move '(northerness and easterness)' to the first mention of aspect.

**Reply**: Thank you. We have implemented the changes as suggested.

L.163-164: 'Height, diameter, and species abundances were measured for species to ensure that vegetational differences evolved through long-term processes and did not reflect the short-term variability of environmental conditions' – On its own, this statement is not correct. In short-lived annuals, these metrics will most certainly reflect short-term variability in environmental conditions. I suspect the plant species living on these rocks to be long-lived perennial species, and therefore one can assume that the height, diameter and abundance of individuals reflect long- term processes and is not solely influenced by short term environmental variability.

**Reply**: We agree with this point. Annuals may have a distinct trait response (height, diameter) and, on La Palma, depended strongly on occasionally occurring precipitation events. However, even if some annual species were found, our plant community is dominated by perennial plants, as you correctly assumed. We, therefore, changed these lines to: " Height, diameter, and species abundances were measured for all vascular plant species. As plant communities were dominated by perennial species, we can expect that vegetational differences evolved through long-term processes and did not reflect the short-term variability of environmental conditions" (L. 206 ff in MS with track-changes).

L.221: 'This makes the greater number on phonolite even more remarkable.' I find this statement a little out of place, given that the results align with your hypothesis, and with the statement that follows.

**Reply**: Yes, the sentence sets up a contradiction that does not exist in this way. We deleted the sentence.

L.223-224: I think this result can be caused by either a lower than expected species richness on the basalts, or a higher than expected species abundance on phonolites, or both.

**Reply**: Thank you for raising this valuable point. We added that it might not only be the unique properties of phonolites enhancing species numbers and plant performance but also effects of basalt that cause these differences between rock types in the discussion. Including this adds another dimension to the discussion.

L.226: I would suggest changing 'colour' to 'temperature'

**Reply**: We see your point that it is not the colour itself but rather rock surface temperature that we discuss as a driver of vegetation differences between basalt and phonolites. However, as we only know for sure that colour differs and we have no measurements for temperature, we prefer to keep the term "colour". However, we added the following explaining term in parenthesis: "...colour (potentially translating into differing rock surface temperature)..."

L.269: I would omit the clause 'though the numbers of endemic species were significantly higher.' I would focus on relative abundance of endemics, which is the relevant metric for the discussion.

**Reply**: Yes, we fully agree with this suggestion and deleted the reference to the number of endemic species. The paragraph is now focused solely on the percentage of endemic species.

L.270: 'and the functioning of phonolites as islands of speciation within a sea of basalt does not seem to apply.' Which makes sense given the geological history of the island?

**Reply**: We agree. The sentence was deleted as we restructured the discussion.

L.271-273: 'However, as most individuals of the typical variety of Cheirolophus junonianus occur on one isolated outcrop and individuals of var. isoplexiphyllus on another one (personal observation), a very local allopatric speciation by adaptive radiation or an ongoing genetic drift could be the underlying cause.' Given the geological history of the island, I would expect the occurrence of these species on one single outcrop to be caused by habitat fragmentation and subsequent extinction in other habitats, rather than speciation on these two outcrops. This is discussed in the subsequent sentence, but the order suggests the authors think speciation is the most likely cause, with which I disagree.

**Reply**: We apologize for the confusion. We agree with you that the occurrences of two *Cheirolophus junonianus* varieties on two separate outcrops are rather the result of habitat fragmentation than small-scale speciation. We rephrased several sentences in the whole paragraph to clarify this point and emphasize that we reject the explanation of small-scaled speciation. The explanation of a burial event of formerly larger and connected phonolite habitats seems more likely. However, we have preserved the order of the two explanations, but we are sure to make the point clearer here with the new sentences (L. 361ff in MS with track-changes):

"As most individuals of the typical variety of *Cheirolophus junonianus* occur on one isolated outcrop and individuals of var. *isoplexiphyllus* on another one (personal observation), it could be suspected that a very local allopatric speciation by adaptive radiation or an ongoing genetic drift could be the underlying cause (Vitales et al., 2014a; 2014b). Thus, the differential geology of phonolites itself does not result in a specialized flora of habitat islands, contrary to e.g., Kruckerberg (1991), and populations of *Cheirolophus junonianus* must be seen in another context. Considering the geologic history of the islands South (Garantje et al., 1998) it is a more possible explanation that this singularity presumes that *Cheirolophus junonianus* belongs to a relict population of plants that were once widely distributed on phonolitic rock before these were covered mainly by basalt. Consequently, lessons learned from other outcrops (Kruckerberg, 1991) cannot be adapted to the phonolitic rocks on La Palma, and the functioning of phonolites as islands of speciation within a matrix of basalt does not seem to apply."

L.293-295: 'Contrasting, plants growing only on phonolites did not experience larger environmental gradients. In accordance, we observed plants on basaltic rocks to be more generalist than plants on phonolitic rocks.' Neither of these aspects are shown in the results, I suggest to either omit these statements or show the data to support them.

**Reply**: This is an observation from the field. Unfortunately, we have no tested data on this. Therefore, we followed the suggestion here and removed the last sentence.

L.305: 'plant growth responses' be consistent in your terminology, this was called reproductive fitness in the rest of the paper.

**Reply**: Thank you. We now use the term "plant growth response" consistently throughout the paper.

L.306: 'of these rocks for the vegetation on these islands which are globally dispersed', I was a bit confused by this sentence. The grammar suggests the islands are globally dispersed, can you mention why is that relevant?

**Reply**: Thank you for raising this point. To simplify the sentence and to give it more clarity, we changed it to: "Higher species numbers and abundances, as well as higher plant fitness, underlines the importance of these rocks for the vegetation on oceanic islands."

L.307: 'Despite the small area covered by phonolites they play a significant role in enhancing plant biodiversity' add 'on the island of La Palma.'

**Reply**: Done, thank you for this comment.

L.307-308: 'Our results contribute to a better understanding of the distribution and plant diversity drivers on islands through exceptional rock outcrops like phonolites' I suggest toning down this statement. The results suggest that these outcrops play a role in the formation of plant diversity on volcanic islands such as La Palma, but the lack of additional data on, in particular, the chemical composition of the different substrates means that your results do not directly contribute to a better understanding of the drivers that lead to this effect.

**Reply**: Thank you. We have deleted the sentence referring to the underlying drivers and rephrased the sentence following your suggestion. It now reads: "Our results suggest that exceptional rock outcrops like phonolites contribute to a better understanding of the formation of plant diversity on volcanic islands such as La Palma."

**Reviewer 2**

**General comments:**

The manuscript is about the effects of parent material (phonolite vs basalt) on plant diversity and performance. The manuscript reads well and is statistically sound, but the key results appear rather

ordinary as one would expect what has been observed: the diversity and plant performance differs between the two parent materials. Although reference is made that the findings are likely to occur also on other islands (or in any regions with and without phonolites), it rather seems of local relevance in its present form. Therefore, the novelty of the study seems limited or needs to be developed further.

A key short coming is the lack of biogeochemical and microclimatic data. Although the study has an interdisciplinary aim and discusses biogeochemical mechanisms to explain the observed plant diversity pattern, it does not include any data on biogeochemistry or soils (e.g. nutrient status, texture, water holding capacity, soil organic matter) or microclimate.

**Reply**: We agree that a holistic study including vegetational and high-resolution biogeochemical data would be best. The manuscript clearly approaches the topic from the disciplines of vegetation sciences rather than biogeochemistry. Nevertheless, this study is the first step into acknowledging the vegetational differences between phonolites and basalt and builds a basis for subsequent studies to thoroughly test the underlying petrographic and biogeochemical conditions responsible for these differences.

To meet the requirements of a paper published in the journal of Biogeosciences we added more literature about the petrographic and biogeochemical characteristics of phonolite and basalt in the text and in an table that offers a comparison between the chemical characteristics of phonolites and basalt (Appendix A1). The additional information strengthens our analysis.

It even appears that one of the rock types contains serpentinite, but although this has fundamental impacts on plants, data are lacking and it remains unclear to which extent serpentinite contributes to phonolite (or basalt).

**Reply**: All our plots could be classified as phonolites or basalts. Even though serpentinite is based on basaltic rocks in the genesis, there was no evidence in the field or the relevant literature on the geology of Palma that they could be serpentinites. The typical mineral compositions that make up serpentinites could not be determined.

An outstanding amount of scientific work has been published about habitat islands that analyse vegetation on serpentinites or gypsum (Chiarucci et al., 1998; Pausas et al. 2003, Harrison et al., 2006; Kazakou et al., 2010). To put our work into context, we used findings from these papers to discuss our results on phonolite.

Moreover, some regions on La Palma are rather dry (no data are given in the manuscript), but the modifying role of parent material on water storage (via texture, color, soil depth etc.) remains unexplored.

**Reply**: Thank you for the comment. We agree that it is helpful for the reader to know more about the climatic details of the studied sites in general. We improved the manuscript by providing some climate data from publicly available climate models (Karger et al. 2017) to give the readers an impression about temperature and precipitation. This will also help to underline that we do not have any soils at the studied plots yet to analyse texture, soil depts and soil water storage.

I am trusting the authors that there are geochemical/site differences, but I am convinced that the reader of Biogeosciences (and me) expects the authors to dig deeper and provide a more quantitative links between plants, microclimate, and biogeochemistry.

**Reply**: We added literature on geochemical difference between phonolites and basalt to strengthen the main points of the paper (e.g., Appendix A2).

Overall, the novelty of the findings appears rather limited and the amount of data shown seem insufficient for a biogeochemical journal. In my eyes, the authors have to provide additional data requiring a rewriting of the manuscript.

**Reply**: Thank you for your assessment of the manuscript and the many helpful comments. Radical editing of the manuscript and the addition of biogeochemical literature have increased the manuscript's quality enormously. We hope to convince you that this study is a first step towards understanding the drivers of vegetational differences between phonolites and basalt and a baseline for further analysis including sound plot-specific data on microenvironmental conditions and chemical compositions.

The current volcanic outbreak of the Cumbre Vieja volcano on the island La Palma underlines the up-to-dateness of our research topic and is a current example of how geology interacts with vegetation. We are convinced that the current event on La Palma will lead to additional interest in our study on vegetation comparison between phonolites and basalt.

**Specific comments:**

Line 19, 20 The 3rd sentences reads trivial; I suggest to combine it with the second one.

**Reply**: The third sentence was combined with the second one following your suggestion and now reads: "A similarity in magmatic origin translates into high global comparability of substrates of volcanic islands on the oceanic crust with, however, slightly chemically or physically differentiated petrography in some places".

Line 21 'traits' is a term that is hardly used by geologists; I would find characteristics more adequate

**Reply**: Thank you. We exchanged the term 'traits' with 'characteristics'.

L. 23 replace 'accumulating' by 'growing'

**Reply**: Done accordingly.

L.37 impacts of geodiversity on biodiversity have long been under-researched – I do agree that it has largely been ignored but 'geobotany' is an established field (e.g. Ellenberg)

**Reply**: Thank you for this comment. We are pleased to read that we have the same position on the fact that the influence of geodiversity on biodiversity has long been ignored. We have rephrased the sentence accordingly. It is also true that H. Ellenberg and others have already discussed and analysed

such connections. Geobotany can encompass these aspects, but as a discipline, it has long been seen as a not clearly defined and broader concept of the geographic distribution of vegetation or the effect of the space itself (Rübel, 1927; Vigo, 1998).

L. 86 somewhat early to start in the 'we-form' in the introduction

**Reply**: Agreed, we changed this sentence into the third person.

L. 90 rephrase the sentence

**Reply**: The sentence was rephrased, divided into two parts, and the information formerly in parenthesis was incorporated into the main text. The passage now reads: "Circulating leachate reaches the rock's surface and evaporates, exposing its dissolved mineral content and enabling the development of secondary mineral assemblages (Spürgin et al., 2019). These can contribute to plant nutrient supply, which is also why ground phonolite rock powder can be used as an effective fertilizer (Faccini et al., 2015)."

L. 126 Hypothese i. Link of plant species richness and growth conditions? This is an interesting one and appears somewhat abrupt and not necessarily the case - it requires some introduction beforehand

**Reply**: The introduction has been changed considerably, and we now find this hypothesis to be better introduced.

L. 139 Methods and information provided are minimal. Please provide climate data and some basic soil (substrate) data (e.g., pH). Climate on La Palma is highly heterogenous and these conditions may also affect vegetation distribution also indirectly via differently textured (=here grain sizes) parent material

**Reply**: We are very sorry, but as mentioned, we cannot provide further information about microclimatic or pH differences between phonolitic and basaltic plots as the fieldwork is already completed. Those plots were chosen in small distances next to each other (a lot have minimum distances of approximately 20 meters), so it is impossible to derive any differences in climate models either (which cannot deal with microclimate). Texture grain sizes could not be investigated since there are no developed soils on the rocks yet.

However, we provided observational climate data from publicly available climate models (Karger et al. 2017) about the study sites in general to support our discussion on the dry climate conditions and circumstances of plant establishment and soil genesis clearer.

L. 232 surface texture – replace texture as for the biogeochemical reader, texture is related to clay, silt and sand.

**Reply**: Agreed, we now use the term "surface characteristics".

L. 235 ff brightness and albedo - any soil temperature measurements available? In current times, records on temperature (and moisture) using in situ loggers are a standard measure in biogeochemistry.

**Reply**: For this study, brightness and albedo measurement are not available due to time constraints. Additionally, this is a low-budget study, and we, therefore, did not have and data loggers at hand.

L. 243ff grain size: how would grain size modify water storage? What was the annual/seasonal precipitation at the sites? Do differences between phonolite and basalt vary between the four sites?

**Reply**: We deleted the term "grain size" to avoid confusion and because it does not add any explanatory value. We furthermore added, as already mentioned, climate data, including precipitation, to the manuscript. However, it is not possible to distinguish between phonolite and basalt based on the precipitation data.

L. 281ff serpentinite. Based on the information given, it is unclear, which of the parent materials contains serpentinite.

**Reply**: We apologize for this confusion. In our study, we only examined the rocks basalt and phonolite, not serpentinite. For more details, see our answer to your second question above.

**Associate Editor:**

Thanks for adressing the reviewer comments. Data on nutrient availability and microclimate are still lacking. For a biogeochemical journal, this is a major shortcoming and strongly limits the novelty of the study. In your response letter, you clearly stated that you will strengthenthe inclusion of geochemical information from the literature, but so far it is not clear how and whether this is sufficient to merit a publication in Biogeosciences. This can only be decided after seeing the written text of the revision.

**Reply:** Thank you very much for your assessment of the manuscript. The inclusion of further literature about the chemical composition of phonolite and basalt directly in the text and a table comparing nutrient availability from these two rock types considerably increase the quality of the manuscript. Although we are approaching the topic from the viewpoint of ecologists, we include a considerable amount of information about geochemical aspects. The current outbreak of the Cumbre Vieja volcano on La Palma shows how closely interrelated geological processes and the biosphere are. We thank you very much for your valuable comments and for considering our manuscript for publication.

---

## Author Response (AR2)

**Author's response**

**Associate Editor**

Dear authors

Thanks a lot revising the manuscript that carefully and please apologize the time until the manuscript was reviewed. Both reviewers noted that the manuscript improved considerably, but their overall judgement differed considerably. While reviewer #1 finds it a valuable contribution and suggested minor revisions with a tendency towards major, reviewer #2 rejected the manuscript due to the lack of biogeochemical data. I also find the manuscript close to the "least publishable unit" for a biogeochemical journal, but above the threshold. Therefore, ask you for major revisions following the comments of reviewer#1.

**Reply:** Thank you very much for your feedback. We highly appreciate your decision to keep our manuscript in the submission system. We carefully revised the manuscript according to the comments and suggestions of reviewer #1.

**Reviewer #1**

I think the authors have made considerable improvements to the manuscript, but I think the manuscript requires additional effort to turn the changes into a cohesive story. Especially the discussion requires a restructure and some re-writing to accommodate the new ideas that have been implemented based on the previous round of reviewer comments. I also have a minor restructure suggestion in the introduction that should be easily addressed and feel that the main aim of the study should be better formulated. These points, and some additional textual suggestions, are specified below.

**Reply:** Thank you very much for your valuable suggestions and comments on the manuscript that we used as guidance to improve our work. Accordingly, the introduction and discussion were adjusted, and further textual changes were implemented.

L.51: "such as e.g.", remove either such as, or e.g

**Reply:** Done.

L.81: "A higher nutrient availability…". This sentence feels disconnected to the rest of the paragraph, which is about the age aspect of phonolites vs basalt. I think you can move this statement to the next paragraph, as the nutrient aspect of phonolites is further explored there.

**Reply:** We started a new paragraph at this point, following your suggestion.

L.115-117: "We aim…La Palma"; I would rewrite and move this paragraph. I think the aim is formulated poorly considering the hypotheses that are formulated later and the data that is

collected (i.e. plant performance is not mentioned and you don't need to measure plant traits to assess species richness and abundance). This paragraph also makes me question the function of the next paragraph, and how this related to the aim presented here. Instead, I think the next paragraph (starting at L.119: "La Palma hosts 159 vascular plant species") connects well to the previous paragraph (ending at L.113: "species specialised to phonolitic rocks"), and I would move the general aim to just before the formulation of the hypotheses.

**Reply:** Thank you very much. We moved this paragraph from the previous position to just before the hypotheses. Now the two paragraphs before are better connected, and we agree that this change increases the quality of the introduction.

L.139: Add "We expect"

**Reply:** Done.

L.229: remove one "by"

**Reply:** Done.

L.230: "reproductive fitness"; remove "reproductive" or change to "performance"

**Reply:** We changed it to "performance".

L.237: "species numbers on phonolites"; add "or both"

**Reply:** We changed it accordingly.

L.237: "studies, which…"; change to "studies that"

**Reply:** We changed the wording according to your suggestion.

L.238: "Reasons…"; only one reason is presented.

**Reply:** We changed your sentence and rephrased it to: "Unrealised niches due to unsaturated evolutionary dynamics in this young and isolated system could be an explanation."

L.242-244: "Characteristics of rocks include, … (as a proxy for rock surface temperature). " I think I would remove these sentences. I don't like how they create an expectation that isn't met in point 3 (while I certainly think it is a valuable discussion point), and they don't capture the paragraph after these 3 points that talks about extinction debt, which is tied to substrate age. Instead, I would encourage the authors to fucus on writing 3-4 strong self-standing paragraphs that convey a single message or explain a single result and don't need an introduction like this.

**Reply:** We deleted these sentences accordingly. We also deleted the numbering of the following paragraphs to avoid listing and to give these points an individual message. Furthermore, we restructured and partly rewrote the following paragraphs to strengthen their message, including your detailed suggestions below.

L.248: "nutrient-rich as it has traditionally been…"; the grounding up of these rocks is not a cause of its nutrient richness, but rather the other way around. Change to: "nutrient-rich, and therefore has traditionally been…"

**Reply:** Thank you for this attentive observation. We changed these sentences substantially due to the restructuring of the whole paragraph. However, we made sure that the content of the according sentence is logical: "Therefore, ground phonolite rock powder has traditionally been used as an inorganic fertilizer (von Wilpert & Lukes, 1998; Ramos et al., 2006; Schoen et al., 2016)."

L.252-254: "Since ground … rock characteristic." Again, the grounding up of these rocks is not a cause of its nutrient richness, but rather the other way around. Please rephrase.

**Reply:** Thank you. Due to the restructuring of this paragraph, we deleted this sentence.

L247-261: The argumentation of this paragraph is not convincing and in places is even incorrect, warranting considerable rewriting, especially given the journal being a biochemical one. The main message that the authors should try to convey is that phonolites and basalts have different chemical compositions, leading to differences in their nutrient availability, which can explain the observed increase in plant performance on phonolitic rocks.

**Reply:** We agree. This paragraph has been substantially rephrased according to your suggestions:

"The rock types phonolite and basalt differ in their chemical composition resulting in different nutrient availability, which explains our observation of increased plant performance on phonolites. Phonolites consist of the potassium-rich nepheline, which dissolves much faster than other potassium sources (Manning, 2010). Various studies indicate that phonolites and the related nepheline syenite contain a higher proportion of potassium than basalt (Manning, 2010; Roqueto do Reis, 2021). Therefore, ground phonolite rock powder has traditionally been used as an inorganic fertilizer (von Wilpert & Lukes, 1998; Ramos et al., 2006; Schoen et al., 2016). Basaltic rock powder has also been used as fertilizer but is considered a less important source of potassium than phonolite (Manning 2010). The usage of ground basalt as fertilizer can also be explained by unclear assignments (potassium-rich trachyte is often assigned to basalt, see Maning (2010)). In addition to geochemical differences, the duration of rock weathering is a decisive factor in providing nutrients. The phonolitic outcrops in Southern La Palma are substantially older than the surrounding basalt, which stems from very young volcanic eruptions (Carracedo et al., 1999). The youngest nearby eruption of the Teneguía volcano took place only 50 years ago, in 1971."

L.265: "studies…" only 1 study is provided.

**Reply:** Thank you, this part has been deleted as we restructured large parts of the discussion.

L.270: "Consequently … unsaturated niches." In the current wording, this refers back to the phonolitic rocks, while these are the old rock type and I would thus expect to have an extinction debt due to recent reductions in habitat area, not unsaturated niches, which I would expect in the surrounding, much younger, basalt habitats. Also see my more elaborate explanation in the previous round of reviewer comments.

**Reply:** We agree that the previous notion was unclear. We deleted the previous sentence, shortened this paragraph and rephrased it from here on:

"In consequence, the species pool in the surrounding basaltic matrix of these rocks is poor. Under the arid conditions of southern La Palma, only very few early successional species establish on these young basaltic outcrops with not more than initial soil formation (Irl et al. 2019). The few rocky outcrops of phonolite are embedded in this species poor matrix of young basalt. We observed partly buried phonolites on which the survival of plants or seedlings during volcanic events was improbable (Garantje et al., 1998). Carracedo et al. (1999) showed that the last phonolite formation occurred in 1585, while basaltic eruptions continue until modern times (Pankhurst et al. 2021). 19 plant species, including *Cheirolophus junonianus,* can solely be encountered on phonolitic rocks (Irl et al., 2015, Muer et al., 2016). This confirms that habitat diversity on islands contributes to their total species richness (Hortal et al., 2009)."

In addition, we added a reference to support the statement of a species poor pool of species in early successional stages on these volcanic substrates.

Irl, S. D. H., Schweiger, A., Hoffmann, S., Beierkuhnlein, H., Pickel. T., and Jentsch, A.: Spatiotemporal dynamics of plant diversity and endemism during primary succession on an oceanic volcanic island. J. Veg. Sci., 30(4), 587-598, 2019.

Additionally, we included a sentence about tthe extinction debt that might explain higher species numbers on phonolites compared to basalt earlier in the discussion (line number in revised manuscript: 244-246).

L.279-280: "Interestingly, … rock type." It is unclear to me what result this statement refers to.

**Reply**: This sentence was replaced by the paragraph mentioned above.

L.299: "While rock chemistry … distinct vegetation." Chemistry and age are brought forth as potential explanations, you do not provide data-driven evidence of a causation. As such please rephrase this sentence.

**Reply:** We rephrased this sentence the following way: "While a diversity of rocks with different chemical characteristics and at different ages supports species richness on volcanic islands, such rock characteristics do not necessarily contribute to higher percentages of endemic plants or compositional distinct vegetation on individual rocky outcrops."

L.300: "Cheirolophus junonianus with its two varieties var. junonianus and var. isoplexiphyllus is confined to phonolites…" I still fail to see how specific mention of this particular species is conducive to answering the research questions. I see the value of mentioning it in the introduction as an example, but not here.

**Reply:** We followed your suggestion, and *Cheirolophus junonianus* is no longer mentioned in this line. The importance of this specific plant species was reduced in the entire discussion.

L.304: "the underlying cause." of what?

**Reply:** We deleted this sentence entirely because the statement is not supported by data but is a subjective observation. This part of the discussion was shortened, and the message was formulated clearer:

"Thus, the differential geology of phonolites itself does not result in a specialized flora. Obviously, the small outcrops of phonolite on La Palma do not suffice to evolve and maintain a substantial set of endemic species, which contrasts with general assumptions that patterns caused by differing topography or discontinuous parent material can be explained by island biogeographic theory (e.g., Kruckerberg 1991)."

L.299-315: The main message of this paragraph is the potential of extinction debt on phonolitic rocks, but I think the authors should drop the one species that occurs only on phonolitic rocks, and instead focus on the larger number of species that occur only on phonolitic rocks compared to the number of species that occur only on basaltic rocks. I also think this can be better tied to the age and geological history of the rock types, as well as the potential for unsaturated niches on basalts. Each of these elements is discussed separately, but I would recommend a restructure to bring at least the extinction debt and the unsaturated niches ideas together on one paragraph.

**Reply:** Thank you for this comment. We also restructured the paragraph considerably. It now focusses purely on endemic species. We moved the sentences about a possible extinction debt to the beginning of the discussion and discussed it with potential unsaturated niches (line number in revised manuscript: 244-246). As extinction debts do not solely apply to endemic species but to species in general, we argue that it fits better into the discussion about different species richness on phonolites and basalt. We hope to convince you that these changes help streamline the discussion and increase the quality of this manuscript.

As suggested, the emphasis of the paragraph is no longer on *Cheirolophus junonianus,* and we now discuss the entire pool of endemic species targeted in our investigation.

L.317-330: I fail to see how the main message of these two (very) short paragraphs relate to the aim of the study.

**Reply:** We deleted the two short paragraphs and instead extended the last paragraph that now contains a stronger reference to our aims. The last paragraph now reads:

"Despite the limited spatial extent of phonolites on La Palma they contribute to insular habitat heterogeneity that translates into increased species richness and abundance as well as higher plant performance. These phenomena are facilitated by the specific characteristics of phonolite rock, like high nutrient availability fortified by longer geological timeframes for erosion compared to basalt. We are not aware of other studies conducted in locations where phonolites can be encountered that explore their potential role as exceptional plant habitat islands, even though phonolites can be found all over the world (Garcia et al., 1986; Ackerman et al., 2015; Hagos et al., 2017). Therefore, further investigation is needed to investigate whether the patterns encountered on La Palma may also be found on comparable phonolitic rocks in other areas of the world. Their benefits for biodiversity found in this study need to be recognized and valued. Especially for isolated areas such as islands, phonolites can contribute to small-scale biodiversity hotspots and our findings suggest that they should be conserved."

---

## Author Response (AR3)

Dear Frank Hagedorn,

Thank you very much for guiding the peer-review process of the paper "Geodiversity and biodiversity on a volcanic island: The role of scattered phonolites for plant diversity and performance" (bg-2021-107). We are delighted that the manuscript was accepted for final publication. The final paper version includes the changed names of tables and figures in the appendix, as described in the acceptance email.

Kind regards on behalf of the authoring team,

Anna Walentowitz